# Social Network Analysis of Agonistic Behaviour and Its Association with Economically Important Traits in Pigs

**DOI:** 10.3390/ani10112123

**Published:** 2020-11-16

**Authors:** Saif Agha, Emma Fàbrega, Raquel Quintanilla, Juan Pablo Sánchez

**Affiliations:** 1Animal Breeding and Genetics, Institute for Food and Agriculture Research and Technology (IRTA), Caldes de Montbui, 08140 Barcelona, Spain; Raquel.Quintanilla@irta.cat (R.Q.); JuanPablo.Sanchez@irta.cat (J.P.S.); 2Animal Production Department, Faculty of Agriculture, Ain Shams University, Shubra Alkhaima, Cairo 11241, Egypt; 3Animal Welfare Program, Institute for Food and Agriculture Research and Technology (IRTA), Monells, 17121 Girona, Spain; emma.fabrega@irta.cat

**Keywords:** welfare, aggressiveness, network centrality, network comparison, feeding behaviour, growth

## Abstract

**Simple Summary:**

Aggression behaviour has several negative consequences on the performance and welfare of pigs. Here, a Social Network Analysis (SNA) approach was employed to (1) identify individual traits that describe the role of each animal in the aggression; (2) investigate the association of these traits with performance and feeding behaviour traits. The study was conducted on 326 Duroc pigs reared in 29 pens. Several individual centrality traits were identified and used to calculate the Social Rank Index. The Dominant, Subordinate, and Isolated animals represented 21.1%, 57.5% and 21.4%, respectively. No significant correlations were observed between out-degree (number of initiated agonistic behaviours) and growth traits, indicating the similarity of growth patterns for dominant and non-dominant animals. Furthermore, out-degree was correlated positively with average daily occupation time (time at the feeder/day) and average daily feeding frequency (number of visits to the feeder/day), but negatively with average daily feeding rate (gr/min). This may indicate the ability of non-dominant pigs to modify their behaviour to obtain their requirements. The Hamming distances between networks showed that there is no common behaviour pattern between pens. In conclusion, SNA showed potential for extracting behaviour traits that could be used to improve pig performance and welfare.

**Abstract:**

Aggression behaviour has several negative consequences on the performance and welfare of pigs. Here, the Social Network Analysis (SNA) approach was employed to (1) identify individual traits that describe the role of each animal in the aggression; (2) investigate the association of these traits with performance and feeding behaviour traits. The study was conducted on 326 Duroc pigs reared in 29 pens. Several individual centrality traits were identified and used to calculate the Social Rank Index. The Dominant, Subordinate, and Isolated animals represented 21.1%, 57.5% and 21.4%, respectively. No significant correlations were observed between out-degree (number of initiated agonistic behaviours) and growth traits, indicating the similarity of growth patterns for dominant and non-dominant animals. Furthermore, out-degree was correlated positively with average daily occupation time (time at the feeder/day) and average daily feeding frequency (number of visits to the feeder/day) but negatively with average daily feeding rate (gr/min). This may indicate the ability of non-dominant pigs to modify their behaviour to obtain their requirements. The Hamming distances between networks showed that there is no common behaviour pattern between pens. In conclusion, SNA showed the potential for extracting behaviour traits that could be used to improve pig performance and welfare.

## 1. Introduction

Mixing unacquainted pigs is a common process in commercial farms. However, this process usually leads to an increase in agonistic behaviour among pigs, who attempt to establish a dominance hierarchy within the group [1]. Such behaviour has several negative consequences on the performance, health and welfare of pigs, e.g., growth, skin lesions [2]. Controlling the agonistic behaviour of pigs is challenging, especially under the current intensive production systems [3]. Getting insights into pig behaviour post-mixing would help in controlling aggression, improve welfare and subsequently increase production efficiency.

Traditional experimental dyadic approaches to studying aggression in pigs have an important limitation as they ignore the social structure of the group [4]. Recently, the application of Social Network Analysis (SNA) in animal behaviour has gained much attention due to its potential to measure the direct and indirect social relationships between the animals [4,5]. The SNA approach provides a great advantage in describing the social relationship at the group level, in addition to providing measurements for each animal based on its role in the interaction [5]. In wild and zoo animal species, SNA has been used in several investigations to characterize the social structure and group dynamics [6,7,8]. However, only few studies have applied SNA to investigate the behaviour of farm animals. In pigs, SNA showed the potential in monitoring the patterns of agonistic behaviour at various age stages [9] and different kinship levels [10]. Furthermore, SNA has provided a more accurate prediction of the long-term aggression within group compared to dyadic trait methods [11]. Nevertheless, a deep understanding of SNA and its applications in pig behaviour requires further investigation [12].

Agonistic behaviour usually occurred around the feeder in pig farms [13], which may have an impact on the feeding behaviour and growth performance of the animals [14]. The development of electronic feeding stations gave the advantage of obtaining accurate individual feed intake measurements along with other variables that allow for defining the feeding behaviour, e.g., the duration of each visit to the feeder for every animal [15,16]. Understanding the association between these feeding behaviours and agonistic behaviour traits could help to optimize welfare in pig commercial farms [17].

The aim of this study was two-fold. First, to gain insight into the agonistic behaviour of pigs and social structure in the pens by identifying behaviour traits that describe the role of each animal in aggression, quantifying the aggression at the pen level, detecting the communities within pen and comparing the pattern of agonistic behaviour between pens. Second, to investigate the association of the individual agonistic behaviour traits identified by SNA with, performance and feeding behaviour traits.

## 2. Material and Methods

### 2.1. Analysed Traits and Data Collection

The study was conducted on 326 Duroc pigs reared at the IRTA pig control farm in Monells, Girona (Spain). During the fattening period, animals were kept in groups of 10–15 growing pigs in 29 pens (20 pens of gilts and nine pens of barrows) under standard management conditions. Pigs were fed ad libitum with automatic feeders, with each pen having an IVOG feeding station (Insentec, Markenesse, The Netherland) with a single-space feeder which identifies the animal that enters the feeder.

Agonistic behaviour records. Pigs were observed using continuous sampling and individually identified by a number on their back [18]. All occurrences of the behavioral traits included in the ethogram were recorded by indicating the number of pigs that initiated or received the agonistic interaction (the Ethogram is shown in Table 1). Each pen was observed for 40 min every day (from 10:00 to 14:30 h) divided in two rounds of 20 min (with a time gap of 5 min in between observations to allow the observer to rest, change pen and let pigs habituate to his/her presence). Although the time constraints limited the opportunity to conduct longer daily observations, the experiment was conducted when pigs were more active in the facilities at this period of the year (February to April); the effect of the period of observation has been included as a limitation of the study in the discussion. Each day of the week, six pens were observed until completing observations of all 29 pens. The order in which the pens were observed was randomized so that at the end of the 16 weeks observation period, all pens had been observed at different times. The pigs were observed from an average age of 65 days (just after mixing in the fattening pens) to 182 days of age (slaughter age), and 6 weeks of observation were spread over that 16 weeks, with a final six observations/pen at an average of 67, 96, 126, 156, 170 and 180 days of age. An agonistic behaviour is defined, in this study, as a physical action started by one pig towards another with a potential negative effect on the receiver and were classified into four types: head knocking, biting, fighting or chasing the opponent. By summarizing all types of agonistic behaviour that occurred between each few animals, a total aggression record was calculated and used as an input for the SNA.

Production performance traits. Based on individual feed intake records, average daily feed consumption in kg/day (ADC) was computed [19]. Animals were weighted every two weeks, and their subcutaneous fat thickness measured by ultrasounds PIGLOG 105(FRONTMATEC-, Kolding, Denmark). Average Daily Gain in kg/day (ADG) was calculated through a within-animal linear regression model of body weight on age, this regression was solved using the function lm() in R base package [20]. This regression was also used to adjust Body Weight at 180 days (BW). Back-fat thickness at 180 days (BF) was also adjusted for each animal based on 4–11 records using also a linear regression model fitted with lm() in R [20]. Food Conversion Ratio (FCR) was calculated by dividing ADC by ADG.

Feeding behaviour traits. The electronic feeders provided measurements for the duration of each visit to the feeder of every animal. Using these measurements, four traits were calculated and defined for every animal: average daily feeding rate (FR, average feed intake per unit of time, g/min), average daily feeding frequency (FF, total number of visits to the feeder per day, in units), average daily occupation time (OT, time at feeder trough per day, in min/day) and average daily time between consecutive visits (FInt, the mean of time between two consecutive visits per day, in min/day) [15]. The association between individual performance, feeding behaviour traits and individual SNA derived traits were calculated using Spearman rank correlation using R software [20].

**Table 1 animals-10-02123-t001:** The ethogram used to record the agonistic behaviour (adapted from Turner et al., 2006 [21]).

Agonistic Behaviour	Definition
Head Knocking	One pig is addressing a headbutt towards another pig at an approximate rate of > = 1 s per 3 s with a negative reaction from the receiver (retreat or attack)
Biting	One pig is addressing an aggressive damaging behaviour (bite) towards another pig at an approximate rate of >1 s per 3 s
Fight	Two pigs are involved in a reciprocal damaging behaviour (including bites and head knocks) an approximate rate of > = 1 s per 3 s. A new fight was recorded after 3 s of non-interchange of damaging behaviours.
Chase	One pig follows another pig at an approximate rate of > = 1 s per 3 s, with a negative reaction of the receiver (retreat or attack)
Initiation of action	
Initiator	Pig which first addresses the agonistic interaction towards the other pig (if not possible to record, then “non-identified” was recorded)
Receiver	Pig which is the recipient of the first agonistic action

### 2.2. Social Network Analysis

SNA was performed using “*igraph*” package in R software [22]. SNA transforms the numerical data of agonistic interaction between animals to graphs, where the animals are displayed in term of nodes. The nodes are connected through edges, i.e., lines, which represent the interaction between animals. The edges can be directed from the animal initiating to the animal receiving the agonistic action and weighted based on the number of times the action was repeated. SNA was used to obtain measures that describe the social behaviour of animals at both individual and group levels. The individual measures give a score for each animal to describe its role in the network. The group measures derived from SNA describe the characteristics of the entire network, e.g., the pen. The group-degree, group-eigenvector, group-betweenness and group-closeness centrality scores at the pen level were calculated and normalized to a value between 0 and 1, following Freeman’s centrality equation [23]. The definition of each individual and group measure is listed in Table 2.

### 2.3. Social Hierarchy

To define the social hierarchy in the Duroc population under study, the animals were classified based on their previously defined degree centrality scores. First, the animals that have an all-degree score equal to zero were classified as “Isolated”. These animals were not involved in any agonistic behaviour with their pen-mates. For the animals that have an all-degree score larger than zero, a Social Rank Index (SI) was calculated by dividing the individual out-degree centrality by its all-degree centrality. The calculation resulted in a score from 0 to 1, which was classified to, “Dominant” for SI ≥ 0.70 and “Subordinate” for SI < 0.70 [27]. Thus, the Dominants attacked more animals and received less attacks, while Subordinates received more attacks than they initiated.

### 2.4. Network Comparison

To compare the behaviour patterns between pens, the Hamming distance was calculated [28,29]. The Hamming distance between two networks is equal to the number of addition/deletion operations needed to transform the edge set of a network into that of the other. However, the Hamming distance can only be calculated on networks with equal number of nodes. Therefore, we only included 19 pens, which were divided into two groups: Group 1 containing 11 pens with 11 animals each, and Group 2 containing eight pens with 12 animals each. For each of these groups, the Hamming distance was computed using *“sna”* package in R software [30], and a distance matrix between each pair of networks/pens was obtained. The multidimensional scaling (MDS) was plotted for each group to visualize the Hamming distances between each pair of networks.

## 3. Results

### 3.1. Agonistic Behaviour, Social Network, and Hierarchy

A total of 551 agonistic interactions were observed in the studied Duroc population. Most of these interactions were head-knocking (74%), followed by bite behaviour (19%), while fight and chase behaviours represented 5% and 2% of actions, respectively. The number of animals and couples involved in agonistic actions for each pen are shown in Appendix A. Total aggression record, i.e., the sum of the actions that occurred between each couple of animals, was used as an input for SNA.

Descriptive statistics for social network individual measures, feeding behaviour and performance traits of Duroc pigs are shown in Table 3. The maximum all-degree centrality was 11, while both in-degree and out-degree showed a maximum of 6. The percentage of each SI category in the Duroc population, which was calculated based on SNA individual centrality scores, showed that Dominant and Subordinate animals represented 21.1% and 57.5% of the Duroc population, respectively, while 21.4% were Isolated animals.

The correlations among SNA individual scores are shown in Table 4. High correlations were observed between all-degree and both out-degree and in-degree centralities, while low correlation was obtained between the out-degree and in-degree centralities (r = 0.36). The correlation between out-degree and closeness was high (r = 0.75). Furthermore, betweenness centrality was highly correlated with all-degree, out-degree and in-degree (r > 0.70).

### 3.2. Association between Individual Agonistic Behaviour, Feeding Behaviour and Production Traits

The correlations between SNA individual scores, and feed efficiency, growth and feeding behaviour traits are shown in Table 5. Out-degree showed a positive but low correlation with FCR. Furthermore, no significant correlations were found between SNA individual scores and ADC, except closeness, which also showed a significant low correlation with FCR. No correlations were observed between out-degree and each of BF, BW and ADG, while closeness showed low positive correlations with BF, BW and ADG. Out-degree was found to be correlated positively with OT and FF, but negatively with FR.

### 3.3. Group Measures and Networks Comparison

Descriptive statistics for network density, reciprocity, group-degree centrality, group-eigenvector centrality, group-betweenness centrality and group-closeness centrality measurements, along with the performance properties for each pen, are shown in Table 6. The network density for the studied Duroc pens ranged between 0.01 and 0.30. The reciprocity ranged from 0 to 0.45. The estimates of the group-eigenvector were higher than 0.27 for all the studied pens. The group-betweenness centrality ranged between 0 and 0.48, and group-closeness ranged between 0.01, and 0.48. The results of the modularity estimates for each of the 29 pens are shown in Figure 1. The modularity estimates found to be positive in 23 out of 29 studied pens, while the remaining six pens showed a zero modularity.

The MDS plot based on the Hamming distances between each pair of networks in the two groups of pens with equal number of animals are shown in Figure 2. The MDS plot showed high distances between most networks, indicating the dissimilarity of the dyadic interactions within different pens.

## 4. Discussion

### 4.1. Agonistic Behaviour and Social Networks

Agonistic behaviour in pigs could be classified into different categories [31]. In this study, head-knocking was the most observed agonistic behaviour (74%), followed by bite behaviour. Similar studies also reported that head-knocking and biting were the most frequent observed aggressive acts between pigs [32,33]. Head-knocking behaviour is considered as a fast, agonistic action that pigs usually use for initiating an attack [34]. Both head-knocking and bite behaviours lead to several injuries in pigs [35]. More severe agonistic behaviours such as fight and chase could lead to important stress and welfare concerns [36], however, they were less frequent in the studied Duroc animals. In any case, reducing all these types of agonistic behaviour is of major interest to pig farmers.

In this context, the SNA showed the potential of obtaining a comprehensive insight into agonistic behaviour of pigs. SNA transformed these records to a network graph representing the social relationships among animals within each pen/group, giving an overview of the agonistic relationships among pen-mates [37]. It can be utilized to identify the animals that are frequently involved in agonistic behaviour and those that avoid any interaction with their pen-mates, i.e., nodes that are not attached to any edges. This could be implemented to manage the aggression in pig commercial farms through the regrouping or removing of animals based on their behaviour.

### 4.2. Social Hierarchy

SNA has the potential to provide measures for each animal that describe its position and influence within the network. Among them, we used the degree centrality scores for identifying the social hierarchy. Büttner et al. (2019) [38] established different dominance indexes to describe the social hierarchy in pigs, and they reported that the centrality score of the animals provided better insights into group structure compared to the dyadic approach. In our study, dominant animals represented about 21% of the studied Duroc pigs. This result agrees with other investigations, which reported that the dominant animals represented 15–31% of pigs [27]. These dominant pigs tend to initiate attacks, i.e., high out-degree, with other pen-mates as they compete for leadership in the pen. On the other hand, the subordinate pigs represented more than half of the animals in the pen (57.5%). This type have lower out-degree and higher in-degree centrality scores compared to dominant pigs. A tendency in the displacement between social rank categories was commonly found between dominant and subordinate pigs [27]. The other type of animal is “isolated”, which had an all-degree score of zero. These animals preferred to hide in a corner in the pen and avoid involvment in any agonistic behaviour [13]. This avoidance strategy is common in pigs, although it may increase the stress level of these animals [39].

The low correlation estimated between out-degree and in-degree (r = 0.36) implies that dominant animals tend to initiate more attacks, i.e., high out-degree, and receive fewer compared to subordinate animals. More interestingly, a high correlation was observed between ranks based on the out-degree and closeness centrality scores. Büttner et al. (2015) [9] also observed high correlations between out-degree and closeness in growing pigs. Closeness centrality illustrates how well an animal is connected to other group members. This correlation suggest that dominant animals tend to be more closely connected to many animals in the pen, which reflect their potential influence on the group [4]. A high correlation was observed between out-degree and betweenness centrality (0.70), which may indicate that dominant animals have a tendency to connect subgroups of animals in aggression. Both the closeness and betweenness centralities are considered as indirect measurements of the influence of individuals on aggression. Taken as a whole, these results suggest that dominant animals used to have a central location in the social network, i.e., pen. These estimates show the potential of SNA to capture the direct and indirect relationships between animals, which allows for the identification of complex social structures.

### 4.3. Association of Agonistic Behaviour with Feeding Behaviour and Production Traits

The agonistic behaviour of pigs is usually observed around the feeder due to competition over resources [13]. Investigating the influence of agonistic behaviour on feeding and production traits is essential to improve the production and welfare of pigs. The low but significant positive correlation between out-degree score and FCR allows for hypothesising that the high energy consumed during aggression may impair the feed efficiency of dominant animals. Thus, agonistic behaviour could be one of the reasons for energy consumption in pigs [40]. However, we should take into account that aggression affects both dominant and non-dominant pigs [41], which may be related to the significant correlation of FCR with all-degree and closeness scores.

On the other hand, a positive correlation between OT and out-degree (but non-significant between OT and in-degree) was observed. According to this, dominant animals occupy the feeder for a longer time compared to their subordinate counterparts. The low correlation between out-degree and ADC suggest, however, that these animals do not consume more feed despite their longer stays in the feeder. Similar studies have also found no differences in ADC between dominant and subordinate pigs [27]. In close agreement, non-significant associations of growth and fat deposition traits (ADG, BW and BF) with both in-degree and out-degree scores were observed, which could be interpreted as an indication of a similar intake and growth patterns for both dominant and non-dominant animals. In line with our results, Campler et al. (2019) [27] reported that female pigs with different social rankings showed similar growing and litter performances when the feeding time and feeding visit length were increased. In our Duroc population, a negative correlation was observed between FR and OT, which suggest that animals that spend less time on the feeder tend to have a higher feeding rate, and that subordinate and isolate pigs are able to modify their feeding behaviour to obtain their feed requirements [42]. This is not unexpected in the ad libitum feeding strategy used during the control, which may give the subordinates and isolated animals a chance to obtain their requirements without being involved in unnecessary agonistic behaviour.

The association between the agonistic behaviour and performance of pigs could be affected by several factors, such as feeding patterns, housing conditions, and group dynamics. Our results support that agonistic behaviour could be associated with the feeding behaviour and growth performance of pigs, but consequences for production performance are expected to be particularly relevant when resources are restricted and/or the time of feeding is limited.

### 4.4. Social Structure in the Pens

Several measurements could be obtained from SNA for describing a group structure [4]. Here, we used the most applicable measurements in the animal behaviour context to understand the behaviour within a pen. Low network density scores (ranging from 0.01 to 0.30) were observed for agonistic behaviour in all studied pens. Low density estimates indicate that only a few animals in each pen were engaged in aggression, while the rest of the animals avoided interaction with other pen-mates. In the same line, only a limited proportion of Duroc pigs seemed to be involved in reciprocal attacks, as denoted by the reciprocity measurement being lower that 0.45 in all pens. Besides, particularly low group-degree centrality scores (below 0.27) were found in the analysed Duroc pens, whereas the group eigenvector centrality scores were found to be particularly high (above 0.69). Taking former results into account, the low density and group-degree centrality along with the high group-eigenvector centrality in the studied pens suggest that the Duroc pens display high aggression behaviour, involving a few animals inside each pen. These findings are in line with results obtained in previous studies conducted with purebred Yorkshire and Yorkshire*Landrace crossbred pigs [11].

Low estimates of group-betweenness and group-closeness centrality scores were found (<0.20) in our Duroc population. Similar results were previously obtained in purebred and crossbred German Landrace and Large White pigs [43]. These results indicate a lack of direct connection between animals in the pen, which could be a consequence of the avoidance strategy that some animals execute to prevent aggression [44]. The subsets of animals within a pen that consistently interact among themselves more frequently compared to other group members are forming communities, which can be identified using the Modularity method [25,45]. In primate animals, chimpanzees appeared to form communities based on agonistic behaviour [46]. In our study, modularity was used to identify the communities in the Duroc pens and summarize the social bonds of its members. Having a range between +1 and −1, the non-zero values of modularity represent a deviation from randomness [26]. Modularity values found to be positive in 23 out of the 29 Duroc pens. Despite the high variation in modularity values, this result suggests that aggressive behaviour used to occur within small communities in most pens. That would support the idea that only a few pigs are interacting among themselves, constructing what could be called a “social hub” [10]. Animals within a social hub are usually competing for leadership. However, forming a social hub post-mixing seems to establish a stable dominance relationship within the pen which, in turn, has the positive effect of reducing aggression [10].

The agonistic behaviour between pigs has been analysed at the pen level. However, it is interesting to explore whether there is a similarity in the pattern of agonistic behaviour that Duroc pigs perform in the different pens. The MDS plot, based on the Hamming distance between networks, revealed an important dissimilarity between most networks, which suggests that there is no clear common pattern of agonistic social interaction between pigs in our Duroc population.

### 4.5. Implication for Pig Commercial Farms

The use of new technology to monitor the behaviour of pigs during the day is expected to be a common practice in commercial farms in the near future [47]. The development of sensors, video cameras and computer software that provide automated data collection will facilitate the implementation of SNA as a tool for management based on a continuous monitoring of the pens to improve welfare. For example, the identification of certain individuals responsible for direct and/or indirect increases in agonistic behaviour can be used for regrouping or removing these individuals. Furthermore, the integration of the agonistic and feeding behaviour information could be used to adjust the housing conditions, location of feeding facilities and the feed distribution frequency during the day to reach an optimal rearing condition for pigs. Furthermore, it would be of considerable value to combine the individual SNA behaviour traits with the genetic information of each animal to develop breeding programs that improve the performance and welfare of pigs simultaneously.

As a limitation of the current study, pigs were only observed every day from 10:00 to 14:30 h. However, the time of the day has been found to influence activity and agonistic patterns in pigs [48]. Therefore, we recommend that future investigations should include the effect of the day’s length on agonistic interactions, considering the modulation of testosterone by melatonin levels. Furthermore, in our study, only females and castrates could be studied, and therefore a lower level of agonistic encounters would be expected compared to having observed entire males. In the EU scenario of increasing pressure to stop piglet castration, further studies using entire males would yield interesting results for the industry.

## 5. Conclusions

Controlling the agonistic behaviour is challenging in pig commercial farms. In the present study, we demonstrated the potential that SNA holds for obtaining insight into pig behaviour and describing the direct and indirect social relationships between animals. Using this approach, we were able to identify behaviour traits that describe the role of each animal, quantify the aggression at the pen level, detect the communities within the pen and compare the pattern of agonistic behaviour between pens. The extraction of new behaviour traits and the integration of them with the performance traits would help in building up future strategies for optimizing management of modern pig commercial farms.

## Figures and Tables

**Figure 1 animals-10-02123-f001:**
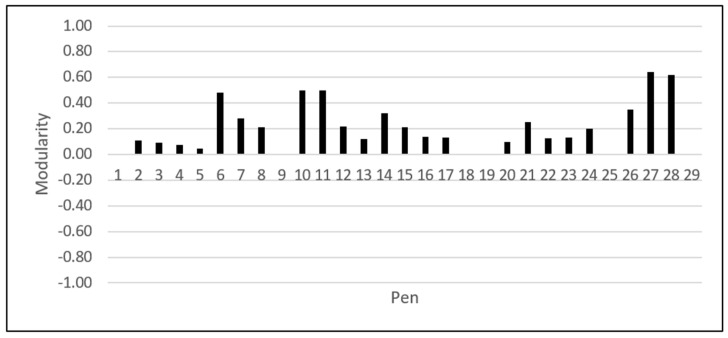
Modularity values estimated for each of the studied 29 Duroc pens.

**Figure 2 animals-10-02123-f002:**
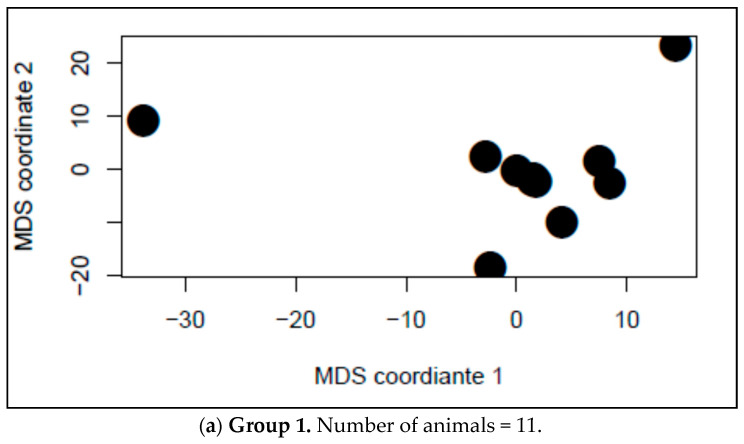
The Multi-Dimensional Scaling (MDS) projection based on the Hamming distances of agonistic behaviour patterns between networks (pens) with the same number of animals, (**a**) Group 1, includes 11 pens with 11 animals each and (**b**) Group 2, includes 8 pens with 12 animals each.

**Table 2 animals-10-02123-t002:** The definition of each Social Network individual and group measures.

Measures	Definition
Individual measures	
All-degree centrality	The number of edges which are attached to a node.
Out-degree centrality	The number of tail-ends adjacent to a node, which represent the number of initiated agonistic behaviours with other pen-mates.
In-degree centrality	The number of head ends adjacent to a node, which represent the number of received agonistic action of the current pig.
Eigenvector centrality	The connectivity of an individual within its network, according to the all-degree centrality of the animal and the all-degree centrality of the animals that connect with.
Betweenness centrality	The number of shortest paths that pass through the considered animal. This measures the importance of the animal in connecting different subgroups the pen in aggression.
Closeness centrality	Measures how close an animal is to all other nodes in a pen, based on the shortest path length between that animal and all other animals in the network.
Group measures	
Network Density	The number of observed edges divided by the number of all possible edges in the network.
Network Reciprocity	Calculates whether the relationships between two animals are reciprocated. The Reciprocity value ranges between 0 and 1. [24].
Modularity	The strength of the division of a network into communities. The Modularity has a value between +1 and −1, where nonzero values indicates the presence of communities within the network, while zero modularity indicates that the division of within-community edges is not different from what would be expected from the randomized network. [25,26].

**Table 3 animals-10-02123-t003:** Descriptive statistics for social network individual measures, feeding behaviour and performance traits of analysed Duroc pigs.

Trait ^1^	Mean ± SD	Max	Min
All-degree	2.34 ± 2.18	11.00	0.00
In-degree	1.17 ± 1.34	6.00	0.00
Out-degree	1.17 ± 1.26	6.00	0.00
Closeness	0.01 ± 0.01	0.05	0.00
Eigenvalue	0.43 ± 0.34	1.00	0.00
Betweenness	4.38 ± 8.78	64.00	0.00
FR	40.61 ± 8.06	65.55	15.28
OT	59.54 ± 9.74	102.64	33.02
FF	10.74 ± 2.90	22.64	5.14
FInt	3.60 ± 0.87	6.65	1.66
ADC	2.34 ± 0.45	3.64	0.97
FCR	3.46 ± 0.35	5.26	2.56
BF	18.52 ± 4.17	28.09	7.67
BW	110.62 ± 12.24	137.88	52.21
ADG	0.81 ± 0.10	1.03	0.23

^1^ FR: average daily feeding rate (gr/min); OT: average daily occupation time (min); FF: average daily feeding frequency (nº visits to feeder); FInt: average daily time between consecutive visits (h); ADC: average daily consumption (kg); FCR: food conversion ratio (kg food/kg growth); BF: backfat thickness (mm); BW: body weight (kg); ADG = average daily gain (kg).

**Table 4 animals-10-02123-t004:** Spearman rank correlations among individual scores for agonistic behaviour from a social network analysis.

	In-degree	Out-degree	Closeness	Eigenvector	Betweenness
All-degree	0.82 **	0.81 **	0.58 **	0.30 **	0.80 **
In-degree		0.36 **	0.24 **	0.27 **	0.72 **
Out-degree			0.75 **	0.23 **	0.70 **
Closeness				0.06	0.52 **
Eigenvector					0.26 **

* *p* < 0.05, ** *p* < 0.01.

**Table 5 animals-10-02123-t005:** Spearman rank correlations of individual agonistic behaviour scores from social network analysis with feed efficiency, growth and feeding behaviour traits.

	ADC	FCR	BF	BW	ADG	FR	OT	FF	FInt
All-degree	0.04	0.17 *	−0.07	−0.04	−0.07	−0.18 **	0.14 *	0.11 *	−0.10
In-degree	−0.02	0.09	−0.1	−0.08	−0.11	−0.11 *	0.08	0.09 *	−0.07
Out-degree	0.07	0.18 *	−0.01	0.01	−0.01	−0.18 *	0.15 *	0.10	−0.11
Closeness	0.15 *	0.17 *	0.12 *	0.09	0.07	−0.06	0.20 *	0.01	−0.05
Eigenvector	−0.06	−0.02	0.01	−0.04	−0.07	−0.02	0.02	0.08 *	−0.02
Betweenness	−0.02	0.1	−0.01	−0.05	−0.08	−0.15 *	0.02	0.11 *	−0.07

* *p* < 0.05, ** *p* < 0.01. ADC = average daily consumption (kg/day); FCR = feed conversion ratio (kg food/kg growth), BF = backfat thickness (mm); BW = body weight (kg); ADG = average daily gain (kg/day), FR = average daily feeding rate (gr/min); OT = average daily occupation time (min); FF = average daily feeding frequency (units); FInt = average daily time between consecutive visits (h).

**Table 6 animals-10-02123-t006:** Descriptive statistics for social network group measures and average pen values for feeding behaviour and performance traits of Duroc pigs.

Trait ^1^	Mean ± SD	Max	Min
Density	0.11 ± 0.07	0.30	0.01
Reciprocity	0.16 ± 0.14	0.45	0.00
group-degree centrality	0.14 ± 0.06	0.27	0.03
group-eigenvector centrality	0.69 ± 0.18	1.00	0.27
group-betweenness centrality	0.21 ± 0.15	0.48	0.00
group-closeness centrality	0.19 ± 0.15	0.48	0.01
Modularity	0.20 ± 0.19	0.64	0.00
FR	40.83 ± 5.22	48.86	31.10
OT	59.71 ± 5.64	69.33	47.09
FF	10.72 ± 1.27	14.55	8.70
FInt	3.60 ± 0.32	4.33	3.16
ADC	2.36 ± 0.36	2.90	1.75
FCR	3.46 ± 0.24	3.94	3.00
ADG	0.80 ± 0.07	0.92	0.66
BF	18.71 ± 2.87	23.73	12.35
BW	110.97 ± 7.94	123.23	94.56

^1^ FR: average daily feeding rate (gr/min); OT: average daily occupation time (min); FF: average daily feeding frequency (no visits to feeder); FInt: average daily time between consecutive visits (h); ADC: average daily consumption (kg); FCR: food conversion ratio (kg food/kg growth); BF: backfat thickness (mm); BW: body weight (kg); ADG = average daily gain (kg).

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
