# Peer review of "Social Network Analysis of Agonistic Behaviour and Its Association with Economically Important Traits in Pigs"

_animals, 2020, doi:10.3390/ani10112123_

Round 1
Reviewer 1 Report
The work is interesting and deserves to be published, I have only minor observations that could improve the understanding of the work.
Line 67. "...electronic feeding stations gave the advantage for obtaining accurate individual feed intake measurements along with other variables that allow defining the feeding behaviour [15, 16]".
It would be useful at this point to provide examples of these other variables.
Line 84. "Direct observations once a week for 12 weeks was performed for all the pens in sessions of approximately four consecutive hours (from 10:00h. to 14:00h)".
Light affects the aggressive behavior of pigs through the modulation of testosterone levels by melatonin. The authors must explain why they chose to monitor pig behavior only in that specified time interval. It would have been interesting to have a longer observation time.
Reviewer 2 Report
Dear authors,
Thank you for submitting this work. I believe this is an important area of research (improving welfare in pigs) and I like the way you have used social network analysis to try and understand relationships between pigs in pens.
Unfortunately I found the manuscript quite confusing and I think a lot more clarity is needed in terms of what the aim of your work is and what the application of your work is. It appears you have tried to take a lof of measurements and this led to a discussion full of acronyms, which was really quite difficult to follow in places.
My main comments would be:
- lots more information required in the methods - currently I am not clear exactly how data was collected - this should be replicable so I think some further information is needed here.
- try and streamline the results - remove things that aren't necessarily useful in the context
Specific comments throughout:
L47/48: a couple of examples of the negative consequences would be good
L53: I don’t think the term ‘global’ is right here – it is just the whole group structure, but it isn’t applicable outside of that group
L63: specifically pig or farm animal behaviour?
L71: this sentence isn’t very clearly written at present – suggest rewriting
L84: which particular sampling method? Is this all occurrence sampling? Did you have a focal pig? If so how did you choose them? Further details needed here on the behaviour obs. How many pens did you observe within that 4hr period? How long did you spend at each pen? How were the pigs identified?
Production performance traits: did you not weigh the pigs? Did you just predict their weight gain using models? it isn't quite clear what was done here, if they weren't weighed then how do you know the models are correct?
Feeding behaviour traits: were the feeders able to identify individual animals? Or were the calculations just created on a per grp basis? How did you calculate individual feed intakes?
I think measures of centrality would probably be more clearly presented in a table – did you use all of these measures in your analysis? could you condense measures?
How did you decide on 0.7 as your dominant/subordinate cut off point? What about pigs that sat ‘in the middle’?
Inappropriate scale on figure 1 (y axis)
What is the benefit of comparing behaviour patterns between pens? I think instead of looking at comparison between groups of pigs it would have been better to look at whether the different negative interactions had the same patterns within grps. E.g. Was it always the larger pigs that were the aggressors? Could it be to do with personality instead?
L253 – is number of subordinate pigs not going to be misrepresented by the way you calculated that?
L278 – was this based on actual pig weights though, or just the models?
There are lots of acronyms in the discussion which are quite hard to remember and to follow – suggest maybe putting these in full where you can
L318: what are the subgroups within a pen? This is the first time I have seen mention of these
L346: if they are dominant pigs is putting them with other pigs not just going to move the issue? And furthermore, if dominant pigs are removed will there not be further fight for dominance among the pigs?
I hope these comments are useful to you and will be helpful to you in any revisions you make.
Reviewer 3 Report
Introduction
- You discuss the issues of mixing unacquainted pigs, based on the intro I had thought this study was conducted post-mixing but it looks like, based on your methods, it revolves only around aggression at the feeder. I would revise the intro to be more focused on this aspect and how that influences welfare of the pigs rather than the mixing aspect. Especially since post-mix fights tend to be resolved within 48 hours but here your study goes on for 12 weeks which typically is not done for post-mix aggression studies.
Methods
The methods are severely lacking in detail and needs more attention. This study is not repeatable due to the missing information
- You say that you did a sampling method as described by Martin and Bateson, but do not mention which one. Is this a focal sampling? Continuous? Scan sampling, if scan what are the intervals? Is this one-zero?
- When did the study start and how long did it run for?
- I would recommend putting average number of pigs plus/minus SD per pen rather than the range
- What were the weights of these pigs? The ages?
- Was this study started post-mixing or after hierarchy had been established? Hence the issue with the intro
- Where is the ethogram? None provided as either a table or as supplementary information so unclear what all the behaviors were that were looked at/analyzed – recommend adding this information in otherwise repeating this study is not feasible
- Its unclear if your experimental unit si the pig or the pen, were these pigs individually marked so observers could tell them apart? If they were marked how so?
- I assume access to water as ad lib as well, but not stated
- Why was observations done only once a week? When once a week, I assume this was done around feeding time with the hours provided but again unclear. I am assuming around feeding time based on the intro.
- Were lesions or body scores looked at? Just out of curiosity.
- How many feeders per pen? Would recommend adding an image of the pen set up to see the feeder
Results
- Figure 1 might look better if the – values on the graph were removed and a stamen about how no pen had a negative modularity was observed, looking at this figure it does not appear that negative values were scored
- Might be formatting, but shouldn’t figure titles be below the figure not above?
- Tables might look better if using mean ± SD rather than 2 separate columns, not necessary though
Discussion
- The first paragraph reads like a methods section rather than a discussion, could be shortened to highlight what this study found rather than re-explaining things that should be in your methods (but are missing, see methods review section)
- What are the recommendations of the authors, I liked the second paragraph but it ends abruptly and would have liked to see recommendations as well as what SNA can do. How might this help with future research/producers with helping decrease agnostic behaviors?
- Again liked the social hierarchy but lacking recommendations, based on isolation approach would you recommend putting barriers into pens to help pigs hide? This has been studied
Overall I like the study/paper but it needs work.
Round 2
Reviewer 2 Report
Dear authors,
Please see my comments (in red) in the attached file. I hope these are useful.

Reviewer 3 Report
Thank you for making the changes, tables look much neater now and methods have been greatly improved. The only thing I might suggest changing now might be changing line 83 to read 20 gilts and 9 barrows, the technical terms. Although for this journal might be unnecessary.
Author Response
We thank the reviewers for their recommendations.
We changed the female and castrates to 20 gilts and 9 barrows as the reviewer recommended